# Naringin Exhibited Therapeutic Effects against DSS-Induced Mice Ulcerative Colitis in Intestinal Barrier–Dependent Manner

**DOI:** 10.3390/molecules26216604

**Published:** 2021-10-31

**Authors:** Ruige Cao, Xing Wu, Hui Guo, Xin Pan, Rong Huang, Gangqiang Wang, Jikai Liu

**Affiliations:** 1School of Pharmaceutical Sciences, South-Central University for Nationalities, Wuhan 430074, China; 2020120554@mail.scuec.edu.cn (R.C.); wuxing1996@hotmail.com (X.W.); 2021110498@mail.scuec.edu.cn (H.G.); px520520@126.com (X.P.); 2Hubei Key Laboratory of Radiation Chemistry and Functional Materials, Non-Power Nuclear Technology Collaborative Innovation Center, School of Nuclear Technology and Chemistry & Biology, Hubei University of Science and Technology, Xianning 437100, China

**Keywords:** ulcerative colitis, naringin, intestinal microbiota, inflammation

## Abstract

Naringin is a kind of multi-source food additive which has been explored broadly for its various biological activities and therapeutic potential. In the present study, the protective effect and mechanism of naringin on dextran sulfate sodium (DSS)-induced ulcerative colitis (UC) in mice were investigated. The results showed that naringin significantly alleviated DSS-induced colitis symptoms, including disease activity index (DAI), colon length shortening, and colon pathological damage. The tissue and serum secretion of inflammatory cytokines, as well as the oxidative stress, were decreased accordingly upon naringin intervention. Naringin also decreased the proteins involved in inflammation and increased the expression of tight junction (TJ) proteins. Moreover, naringin increased the relative abundance of *Firmicutes*/*Bacteroides* and reduced the content of *Proteobacteria* to improve the intestinal flora disorder caused by DSS, which promotes the intestinal health of mice. It was concluded that naringin can significantly ameliorate the pathogenic symptoms of UC through inhibiting inflammatory response and regulating intestinal microbiota, which might be a promising natural therapeutic agent for the dietary treatment of UC and the improvement of intestinal symbiosis.

## 1. Introduction

Inflammatory bowel disease (IBD) is characterized as an uncontrollable, nonspecific chronic immune-mediated intestinal inflammation which can be divided into Crohn’s disease (CD) and ulcerative colitis (UC). The UC lesions mainly occur in the colon and the rectal mucosa and sub-mucosa, which are manifested with a persistent diffusion pattern. In recent years, the global incidence of UC has been increasing dramatically, especially in urban areas [1]. Common clinical symptoms of UC include gastroenteritis, fever, diarrhea, rectal bleeding and fecal mucus [2]. Additionally, patients with UC are at higher risk of developing *Escherichia coli*-associated colorectal cancer [3]. 

Currently, the pathogenesis of UC is relatively complicated and has not yet been fully understood. A general consensus exists that UC is closely associated with immune system dysfunction, intestinal mucosal injury, and dysbacteriosis in people with congenital susceptibility [4]. In this regard, the increases of inflammatory cytokines and oxidative stress are known as important incentives to the occurrence and development of UC [5]. Furthermore, extensive studies elucidated the key role of the gut microbiota in the pathogenesis of UC, which postulated that intestinal flora disorders and the occurrence of enteritis also have a close relationship [6,7,8]. Compositional and metabolic imbalance in the intestinal microbiota could change the intestinal mucosal permeability, concomitantly with the infiltrations of pathogenic bacteria and toxins, leading to long-term inflammatory stimulation and even canceration of the colon mucosa [9,10]. Clinical trails revealed that UC can respond to antibiotic therapy, which is consistent with the conception that intestinal bacteria contribute to the inflammation [11,12]. Additionally, increased paracellular permeability and abnormal structures of colonic TJ proteins have been observed in UC patients, leading to the destruction of the intestinal barrier [13]. Therefore, regulating intestinal flora can effectively protect the gastrointestinal tract and promote intestinal health [14]. Notably, more studies will be needed to define host–microbial relationships relevant to UC and amenable to therapeutic interventions.

At present, the main drugs commonly used in the treatment of ulcerative colitis are anti-inflammatories (5-aminosalicylic acid and corticosteroids) and immunosuppressants (infliximab and adamuzumab) [15], but these drugs often have certain adverse reactions and limitations [16,17]. In addition, several studies have indicated that the administration of prebiotics can rebalance gut microbiota in UC patients [18]. Therefore, food-assisted treatment of ulcerative colitis has attracted worldwide attention. Dietary polyphenols may protect the gut by suppressing inflammatory responses and have therapeutic effects on UC [19]. Dietary fiber supplementation can also help maintain the integrity of the colonic mucosal barrier during intestinal digestion [20]. Existing studies have shown that edible berries, such as red raspberry, Goji berry and Maqui berry, can treat ulcerative colitis through their anti-inflammatory effects and gut microbiota regulations [21,22,23]. Naringin is an important flavonoid existing in the fruit or peel of rutaceae grapefruit which contributes to the bitter taste of citrus juices [24]. Naringin is well-known for a vast array of biological activities and immense therapeutic roles, and has been recognized for its anti-cancer efficacy [25], antimicrobial potential, anti-inflammatory activity [26], anti-ulcer role, osteoporosis-reducing exercises, and numerous significant hepatoprotective, cardioprotective, and renoprotective properties [27,28]. A recent article reported that naringin displayed a protective effect on UC in mice [29]; however, the effects of naringin on the balance of intestinal microbiota have not been fully investigated, although it is the major feature of UC.

In this study, the ulcerative colitis model induced by dextran sodium sulfate (DSS) was adopted, and the therapeutic effect and possible mechanism of naringin on UC was explored in terms of regulating intestinal microbiota, restoring the intestinal barrier, and inhibiting inflammation, which provided a scientific basis for further expanding the application of naringin in the later period.

## 2. Results and Discussions

### 2.1. Naringin Ameliorated DSS-Induced UC

UC is an idiopathic chronic inflammatory bowel disease, the mechanism of which is often explored in DSS-induced colitis models [30]. Consistent with human ulcerative colitis, DSS-induced colitis can also cause symptoms such as bleeding in stool, diarrhea, and weight loss [31]. In our study, the body weights of mice were significantly reduced after DSS induction (Figure 1A). As an acknowledged index to evaluate ulcerative colitis, the DAI scores of the DSS group were dramatically increased (*p* < 0.01), however, the scores gradually decreased and UC symptoms alleviated upon naringin treatment in a dose-dependent manner (Figure 1B). Additionally, the length and weight of the colons in each group were also measured (Figure 1C–E). The length (*p* < 0.001) and weight (*p* < 0.01) of the colons significantly reduced in the model group. Naringin administration significantly ameliorated DSS-induced UC, as evidenced by the marked restoration of weight loss and relief of colon shortening in the medium-dose group (*p* < 0.05) and high-dose group (*p* < 0.001).

Inflammatory changes in colitis are limited to the large intestine and are commonly characterized by crypt distortion and goblet cell loss. Histopathological results showed that the control mice had a large number of goblet cells and an intact mucus layer, while those goblet cells and mucus layer in the DSS group were seriously destroyed with extensive infiltration of inflammatory cells (Figure 2A). Such an opened mucus layer might directly contact intestinal pathogenic microorganisms and further accelerate the damage of the mucus layer and inflammatory response in colitis. As expected, the phenomenon was obviously improved by the treatment of naringin, which could reduce the colon injury and restore the integrity of the colon structure. Histopathological scores of each group also demonstrated that naringin intervention could considerably improve morphological structure of colon tissue at medium (*p* < 0.05) and high (*p* < 0.01) doses of naringin therapy. Only the Nar (100 mg/kg) group did not cause colitis symptoms, including DAI scores, colon shortening and histopathological changes. Both DAI score and histological analysis showed that naringin had a significant alleviation effect on colitis.

### 2.2. Inhibition of Proinflammatory Cytokines

Overproduction of pro-inflammatory cytokines is considered as the major factor in the occurrence and development of UC. The complex interaction between infiltration of inflammatory cells and release of inflammatory cytokines could exacerbate the intestinal inflammation [32,33]. Naringin was reported to have potential anti-inflammatory activity [34]. Based on these, the reduction of inflammatory cytokines at colon tissue level and serum level were fully evaluated in Figure 3. Tumor necrosis factor-α (TNF-α), interleukin-6 (IL-6) and interleukin-1β (IL-1β) in mouse colon tissue were detected by immunofluorescence (Figure 3A). The results showed that the secretion of all three factors was significantly increased in the DSS group, and the colon morphology was also damaged. But after naringin treatment, the level of those cytokines decreased and the colonic morphology recovered to some extent. Simultaneously, the contents of TNF-α and IL-6 in mice serum (Figure 3B,C) also dramatically decreased after medium (*p* < 0.05) and high (*p* < 0.01) doses of naringin therapy in comparison with those of the model group through enzyme-linked immunosorbent assay (ELISA). These results suggest that the naringin could reduce inflammation and decrease intestinal permeability.

### 2.3. Regulation of Oxidative Stress Levels

It is well-known that disruption of the mucosal immune barrier causes the accumulation of reactive oxygen species (ROS), resulting in the induction of UC [35,36]. Mohamed, E. A et al. reported that naringin has an antiulcer effect and can regulate the level of oxidative stress. [37]. To further understand whether naringin treatment has an antioxidant effect on DSS-induced UC, three indicators related to oxidative stress were examined in Figure 4. The results show that malonaldehyde (MDA) contents were significantly increased in the DSS-treated group (*p* < 0.001), generally accompanied by the decrease of glutathione (GSH) (*p* < 0.001) and superoxide dismutase (SOD) (*p* < 0.01) levels. By contrast, MDA contents decreased (*p* < 0.05 for 25 mg/kg, *p* < 0.001 for 50 and 100 mg/kg) while GSH (*p* < 0.01 for 25 mg/kg, *p* < 0.05 for 50 mg/kg and *p* < 0.001 for 100 mg/kg) and SOD (*p* < 0.01 for 25 mg/kg, *p* < 0.001 for 50 and 100 mg/kg) contents increased after treatment with naringin in a dose-dependent manner. The contents of the three indexes in the Nar (100 mg/kg) group were similar to those in the control group. These results suggested that naringin could effectively improve the oxidative stress of colon segments.

### 2.4. Expression of Proteins Involved in Inflammation and Intestinal Barrier

An increasing number of studies suggest that the occurrence of intestinal inflammation impairs the integrity of the intestinal epithelial barrier and destroys the dynamic balance of intestinal microbiota [38]. The previous H&E result in our study displayed pathological changes in the colon tissue of DSS-induced colitis, which may also impair the epithelial TJ proteins. In this regard, the expression of proteins involved in inflammation and the intestinal barrier was further investigated through western blotting in Figure 5. The results showed a significant increase in the protein expression of both inducible nitric-oxide synthase (iNOS, *p* < 0.05) and cyclooxygenase-2 (COX-2, *p* < 0.001) in the model group. However, the up-regulations of iNOS (*p* < 0.05) and COX-2 (*p* < 0.001) were almost suppressed by the naringin treatment at the concentration of 100 mg/kg, which was consistent with the results of inflammatory cytokines expression in vivo. Concurrently, the contents of the TJ proteins ZO-1 and occludin in colon tissue was significantly reduced in the DSS group (*p* < 0.05), while the expression of those two proteins increased after naringin intervention (*p* < 0.05 for 50 mg/kg), suggesting the protective effect of naringin on the intestinal tract.

iNOS and COX-2 are considered to be important pro-inflammatory mediators during intestinal inflammation and their up-regulation is associated with transcriptional activation. Up-regulation of iNOS and COX-2 in macrophages and subsequent formations of NO and PEG2 are very crucial in the immune response to infectious agents [39]. Relevant reports showed that iNOS and COX-2 protein up-regulation is one of the characteristic indicators of colitis recovery [40]. Intestinal TJ protein is an important pathogenic factor of intestinal inflammation [41]. Occludin and ZO-1 are representative TJ proteins that play an important role in maintaining the integrity of the intestinal mucosal barrier. These proteins regulate the permeability of the intestinal barrier to prevent the entry of pathogenic antigens into the lamina propria of mucosa and to reduce inflammation and immune responses in vivo [42]. The therapeutic effect of naringin on UC is mainly achieved by reducing the damage of TJ protein and maintaining the structure of the colonic barrier system [43,44]. These results indicated that naringin could effectively improve intestinal epithelial barrier dysfunction and protect UC mice by reducing the inflammatory response and increasing the expression of TJs-related proteins in the intestine. 

### 2.5. Effect of Naringin Treatment on Intestinal Flora of Mice with Ulcerative Colitis

In order to further explore the mechanism of naringin in the treatment of ulcerative colitis, the intestinal flora of mice was studied. In this experiment, fecal samples from the control group, DSS group (Model group) and Nar (100 mg/kg) + DSS group (Nar group) were selected as the research objects, and each group was repeated for five times. The data were analyzed on the online platform of MGI. Operational taxonomic units (OTUs) in the control, DSS, and Nar groups were 57,773 ± 3473.01, 53,794 ± 4198.90, 50,151 ± 3326.51, respectively.

The rank-abundance curve was used to explain two aspects of diversity, which were species richness and community evenness. In Figure 6A, the curve of the control group declines gently and extends far, indicating high species diversity. However, in the DSS group and the Nar group, species diversity and richness decreased. The Shannon index of the OTU level was set as the vertical axis in the rarefaction curve. The curve tends to flatten, indicating that the sample sequencing volume is sufficient, and no more OTU can be found even with the increase of data (Figure 6B). Pan analysis showed that the DSS group had the least number of species, while the control group had the largest number of species (Appendix A). Alpha diversity refers to the diversity within a specific region or ecosystem. As shown in Appendix A, the diversity of intestinal flora in the DSS group decreased, but there was no significant difference between each group. All the above results indicated that the sequencing sample size of this study was sufficient and the sample species covered a wide range.

The Venn diagram can be used to calculate the number of sample shares or unique OTUs (Figure 6C). The histograms illustrating the gut microbiota community structure showed the microbial species and their relative abundance at the phylum level (Figure 6D). There were nine phyla with relative abundance greater than 0.01% in all the three groups, among which *Firmicutes* and *Bacteroidota* were the dominant ones. Compared with the control group, *Firmicutes* decreased and *Bacteroidota* increased after DSS modeling. Upon naringin treatment, *Bacteroidota* decreased while *Firmicutes* increased, respectively. Similarly, a heatmap of the bacterial community of phylum horizontal abundance showed the content changes of these bacterial communities in each sample (Figure 6E). These indicated that naringin could restore *Firmicutes/Bacteroidetes* (F/B) ratios and regulate the structure of intestinal microbiota, which is consistent with the previous study [45]. The Circos graph also confirmed that the common intestinal flora of these three groups mainly include *Firmicutes, Bacteroidota, Campilobacterota,*
*Proteobacteria, Desulfobacte* and *Actinobacteriota* (Figure 6F). It has been reported that *Firmicutes, Bacteroidetes, Actinobacteria,* and *Proteobacteria* account for more than 98% of the intestinal flora. Among them, the increase of *Proteobacteria* reflects the disorder of intestinal flora [46,47]. According to the results, the level of *Proteobacteria* in DSS group significantly increased compared to the control group, and subsequently decreased upon naringin treatment. This further demonstrates that naringin could improve intestinal flora imbalance. Besides, *Deferribacteria* were only found in the control group and the Nar group, while *Cyanobacteria* was only detected in the model group and the Nar group.

Sample comparative analysis enables better comparison of the colony structures of different samples. According to hierarchical clustering analysis of the beta diversity distance matrix, the sample hierarchical clustering tree was constructed. It was found that the control group had a good clustering situation, but the colony was significantly different from that of the model group and Nar group, indicating that DSS could affect the intestinal flora of mice (Figure 7A). Principal component analysis (PCA) in Figure 7B showed that PCA points in the control group and model group were well separated, suggesting the significant differences between these two groups. However, the Nar group was close to the control group, indicating that naringin intervention could improve and adjust the structure of intestinal flora. Additionally, non-metric multidimensional scaling (NMDS) analysis can be used to simplify the research objects of multidimensional space to low-dimensional space for positioning, analysis and classification, which is applicable when exact similarity or dissimilarity data between research objects cannot be obtained. The results of NMDS analysis showed that the distance operation of all values in the control group was similar except for a few discrete values, but the distance between DSS group and Nar group was significant (Appendix A). Moreover, principal co-ordinates (PCoA) analysis can also be utilized to study the similarity or difference of sample community composition. It can be observed from Appendix A that the model group differed greatly from the control group, but approached the control group after naringin administration.

Species differences in the samples at the phylum level were analyzed using the Kruskal-Wallis H test. Compared with the control group, the intestinal microflora of DSS-induced colitis mice (Model group) showed fewer bacterial species, mainly including *Fimicutes*, *Campilobacterota* (*p* < 0.05), *Deferribacterota* (*p* < 0.05), *Verrucomicrobiota* and *Elusimicrobiota*. Meanwhile, the increased bacteria species included *Bacteroidota*, *Proteobacteria*, *Actinobacteria*, *Patescibacteria* (*p* < 0.01) and *Cyanobacteria* (*p* < 0.05). In the Nar group, increased *Bacteroidota*, *Proteobacteria*, *Patescibacteria* and *Cyanobacteria*, and decreased *Fimicutes*, *Campilobacterota*, *Deferribacterota* and *Verrucomicrobiota* were observed (Figure 7C). LEfse analysis results showed that there were significant differences among the three groups in 4 phyla, 6 classes, 14 orders, 20 families and 40 genera. The relative abundance of *Rikenellaceae* increased in the model group, while the relative abundance of *Faecalibaculum* increased after naringin treatment. (Figure 7D and Appendix A).

Dysregulation of intestinal flora also plays an important role in the pathogenesis of ulcerative colitis [48]. The imbalance of intestinal flora has been found to cause intestinal inflammation, and the inflammatory state leads to the imbalance of intestinal flora by changing the intestinal oxidative and metabolic environment [49]. It has been reported that the richness and diversity of intestinal microflora were decreased in DSS-induced UC mice, which was consistent with the results of this study [50]. *Firmicutes* and *Bacteroidetes* are the main components of the complex composition of intestinal microflora [51]. The decline of beneficial bacteria and an increase in pathogenic bacteria induces an intestinal microbiota imbalance in UC, which is characterized by a reduction of bacterial diversity and an decrease in the ratio of *Firmicutes/Bacteroidetes* [52]. Compared with the control group, the relative abundance of *Firmicutes* decreased and *Bacteroidetes* increased in the mouse feces induced by DSS. After naringin treatment, the F/B ratio significantly recovered. Moreover, this study found that *Deferribacteria* were detected in the control group and the Nar group, while *Cyanobacteria* was detected in the DSS group and the Nar group. This may be related to changes in intestinal microenvironment, which provides a direction for further exploration of the influence of naringin on intestinal flora.

## 3. Materials and Methods

### 3.1. Chemicals

DSS (molecular weight 36−50 kDa) was purchased from Aladdin–Holdings Group (Shanghai, China). Naringin (purity > 98%) were provided by Energy Chemical (Shanghai, China). The ELISA kits of IL-6 and TNF-α, superoxide dismutase (SOD), glutathione (GSH) and malondialdehyde (MDA) assay kits were obtained from Nanjing Jiancheng Bioengineering Institute (Nanjing, China). Rabbit polyclonal antibodies for TNF-α, IL-6, IL-1β, COX-2, iNOS, Occludin, ZO-1, goat anti-rabbit IgG secondary antibodies and Texas Red conjugated goat anti-rabbit antibody were obtained from ABclonal Technology (Wuhan, China). Rabbit polyclonal antibodies GAPDH was purchased from Cell Signaling Technology (Boston, MA, USA). All other reagents were commercially available and of reagent grade.

### 3.2. Ethics Statement

Male C57BL/6 mice (18–20 g, 4–5 weeks) were purchased from Liaoning Changsheng Biological Experimental Animal Center (Certificate SCXK 2020-0001; Benxi, China). Seven days before the experiment, mice were maintained under standard specific pathogen-free conditions and given sufficient food and water at 24 ± 1 °C to adapt to the new environment. Animal use and care were approved by the Animal Ethics Committee of South-Central University for Nationalities (SYXK (Wuhan) 2016-0089, No. 2021-SCUEC-AEC-033).

### 3.3. Treatment of DSS Induced Colitis in Mice with Naringin

Mice were randomly divided into six groups (*n* = 10): Control group, Model group, Nar (25, 50 and 100 mg/kg) + DSS groups, Nar (100 mg/kg) group. From day one to day seven, mice were only given normal saline via oral gavage in control group, and the Model group was given DSS (8 g/kg, 0.6 g/mL) in the same way. In Nar (25, 50 and 100 mg/kg) + DSS groups, mice were given the same amount of DSS orally in the morning and a specific dose of naringin in the afternoon. In addition, mice were also treated with naringin (100 mg/kg) only during the experiment for the Nar group. Weight and feces of mice were recorded every day. The DAI data are presented as an average score of the body weight change, stool consistency and hemoccult bleeding, under the guidance of a previously established scoring system [53]. A week later, mice were anesthetized and sacrificed, then the colons, feces and blood were collected for further analysis.

### 3.4. Histopathological Evaluation and Immunofluorescence Analysis of Colon Tissue

The fragments of the mice’s colon were fixed with 10% formalin solution, embedded in the paraffin, sectioned, then stained with hematoxylin and eosin (H&E) for histopathological evaluation according to scoring criteria [54].

Three colon samples from each group were randomly selected and stained for TNF-α, IL-1β, and IL-6. The frozen colon sample was sliced, air-dried, fixed with 4% paraformaldehyde and washed with cold PBS for times. The samples were blocked with 1:10 normal goat serum and incubated with rabbit polyclonal antibody TNF-α, IL-6 and IL-1β for 1h at room temperature. The samples were washed with cold PBS for three times and incubated at room temperature with Texas Red conjugated goat anti-rabbit antibody for 50 min. After that, the samples were washed with PBS three times again. The sample fragments were then dropped with DAPI and incubated in the dark for two minutes to dye the nuclei, washed with PBS three times, dehydrated with ascending ethanol series, and analyzed using a Nikon confocal laser scanning microscope. Photomicrographs were recorded at a 200× magnification.

### 3.5. Inflammatory Cytokines Assay

The serum was separated from the collected blood and the levels of inflammatory cytokines, TNF-α and IL-6, were measured by ELISA kits according to the manufacturer’s protocol. Briefly, the collected serum with appropriate dilution was added to ELISA plate and incubated for 2 h. After washing as required, biotinylated detection antibody was added and incubated for 1 h. After washing the plate, streptavidin-horseradish peroxidase (Sav- HRP) was added for 30 min. The plate was then washed and color substrate solution was added for 30 min away from light. Finally, the reaction was terminated with 10% (*v*/*v*) H2SO4 and the plate was read at 450 nm on a microplate reader (Tecan Spark 10M, Männedorf, Switzerland). The standard curve was constructed with TNF-α (0–1000 pg/mL) and IL-6 (0–1000 pg/mL).

### 3.6. SOD, GSH and MDA Assay

Colon tissue was cut along the midline and homogenized at 4 °C with a high-speed tissue grinder. The remaining homogenate was centrifuged at 3500 rpm, 4 °C for 10 min. Part of the supernatant was carefully taken for quantitative determination of protein concentration, then the contents of SOD, MDA and GSH were determined step by step according to the experimental instructions of related kits.

### 3.7. Western Blot Analysis

The colon segments were homogenized in ice-cold lysis buffer plus the protease inhibitor. The pyrolysis solution was centrifuged at 12,000 rpm, 4 °C for 10 min to obtain the supernatant protein. The extracted proteins were separated by an SDS-polyacrylamide gel electrophoresis and transferred to the PVDF membrane which was then blocked with 5% nonfat milk for 1h. Upon incubation with primary antibodies for COX-2, iNOS, Occludin, ZO-1, and GAPDH overnight at 4 °C and with a strict wash, the PVDF membrane was then incubated with specific goat anti-rabbit IgG secondary antibodies for 1 h at room temperature. After repeated wash, chemiluminescent signals of the protein bands were detected using ECL solution and acquired by the imaging system (ChemiDoc XRS+, Bio-Rad, Hercules, CA, USA). The band scanning densitometry of each protein was quantified through Image J.

### 3.8. Feces Microflora Analysis

Microbial genomic DNA of fecal samples was extracted. Five samples were randomly selected from each group for further study. Specific primers (forward primer: 5′-ACTCCTACGGG AGGCAGCAG-3′; reverse primer: 5′-GGACTACHVG GGTWTCTAAT-3′) were used to amplify the V3–V4 region of the bacterial 16S rRNA gene. The composition of the gut microbiota was then determined by dual-indexing amplification and sequencing approach on the Illumina MiSeq platform followed by USEARCH (version 7.0) bioinformatics analysis.

### 3.9. Statistical Analysis

The results were expressed as the mean ± standard deviation (SD) of at least three independent experiments, unless stated otherwise. Data were analyzed by GraphPad Prism 8 software (GraphPad Software Inc., San Diego, CA, USA). Multiple comparisons were performed using one-way analysis of variance (ANOVA) followed by Tukey’s post hoc test, and statistical significance was set at *p* < 0.05.

## 4. Conclusions

In this study, a mouse model of ulcerative colitis was established with DSS, which showed that naringin had obvious therapeutic effects on ulcerative colitis. Naringin not only significantly improved the weight loss of mice, reduced DAI score and alleviated colonic tissue damage, but also effectively alleviated oxidative stress and reduced the production of inflammatory cytokines. Further investigation found that the mechanism of naringin in treating UC was closely related to that of mucosal TJ proteins. Additionally, analysis on the intestinal microflora of mice showed that the intake of naringin could partially restore the intestinal biodiversity in the DSS-induced UC mice and promote the restoration of the structure of intestinal microbiota. Overall, naringin treatment can significantly relieve ulcerative colitis induced by DSS in mice, and is expected to become a drug candidate for the prevention and treatment of inflammatory bowel disease.

## Figures and Tables

**Figure 1 molecules-26-06604-f001:**
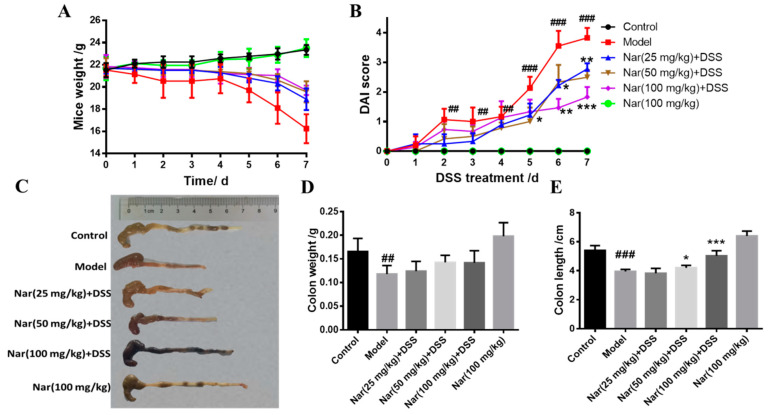
Naringin could alleviate the symptoms of DSS-induced colitis in mice. (**A**) Changes in body weight of mice during the experiment. (**B**) DAI score. (**C**) Appearances of the colon tissues. (**D**) Weights of colon in different groups. (**E**) Lengths of colon in different groups. Data are presented as means ± SD (*n* = 10 per group). ## *p* < 0.01 and ### *p* < 0.001, compared with the control group; * *p* < 0.05, ** *p* < 0.01 and *** *p* < 0.001, compared with the model group.

**Figure 2 molecules-26-06604-f002:**
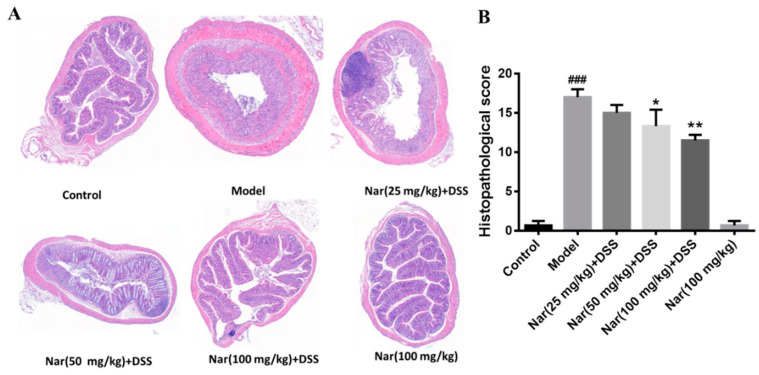
Histopathological changes of colons in ulcerative colitis mice were improved by naringin. (**A**) The colons of each group were processed for histological evaluation (H&E staining, 50×). (**B**) Histopathological scores of each group were evaluated. Data are presented as means ± SD (*n* = 6). ### *p* < 0.001, compared with the control group; * *p* < 0.05 and ** *p* < 0.01, compared with the DSS model group.

**Figure 3 molecules-26-06604-f003:**
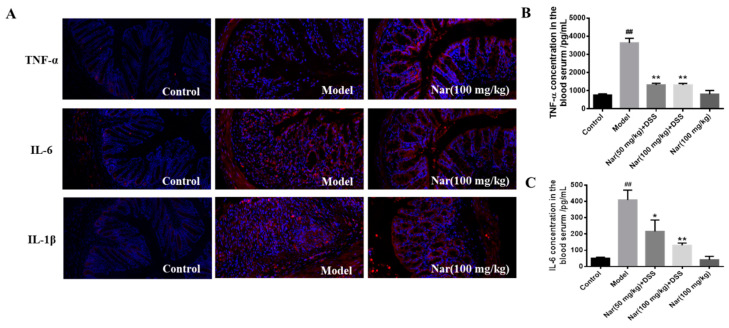
Effect of naringin on the expression of inflammatory cytokines in ulcerative colitis mice. (**A**) TNF-α, IL-6 and IL-1β distribution in the colons of each group (immunofluorescence analysis, 200×). (**B**) Concentration of TNF-α in the blood serum. (**C**) Concentration of IL-6 in the blood serum. Data are presented as means ± SD (*n* = 3). ## *p* < 0.01, compared with the control group; * *p* < 0.05 and ** *p* < 0.01, compared with the model group.

**Figure 4 molecules-26-06604-f004:**
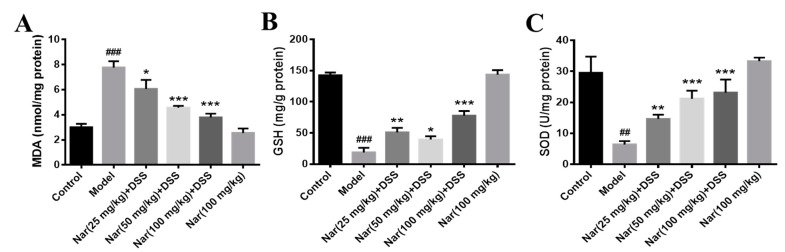
Effects of naringin on oxidative stress levels in mice with colitis. The level of MAD (**A**), GSH (**B**) and SOD (**C**) were detected in the colon tissues of each group. Data are presented as means ± SD (*n* = 3). ## *p* < 0.01 and ### *p* < 0.001, compared with the control group. * *p* < 0.05, ** *p* < 0.01 and *** *p* < 0.001, compared with the model group.

**Figure 5 molecules-26-06604-f005:**
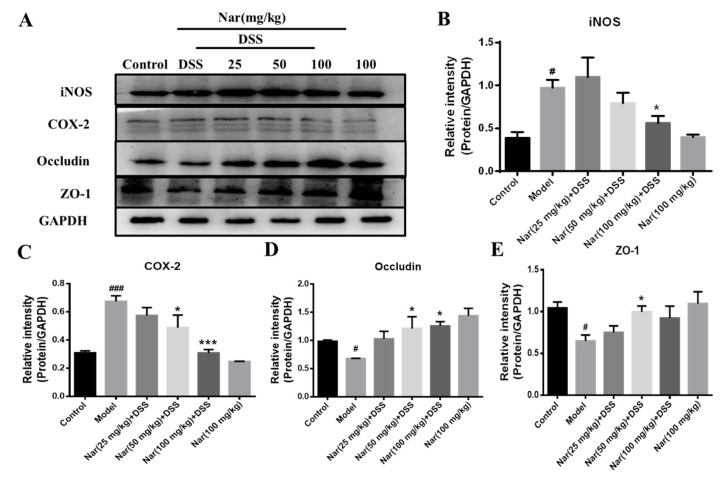
Expression of related proteins in tissues of ulcerative colitis mice. (**A**) The protein expressions of iNOS, COX-2, Occludin and ZO-1 were detected by Western blotting. GAPDH was used as an internal control. The gray density scanning analysis of iNOS (**B**), COX-2 (**C**) and occludin (**D**) and ZO-1 (**E**). Data are presented as means ± SD (*n* = 3). # *p* < 0.05 and ### *p* < 0.001, compared with the control group; * *p* < 0.05 and *** *p* < 0.001, compared with the model group.

**Figure 6 molecules-26-06604-f006:**
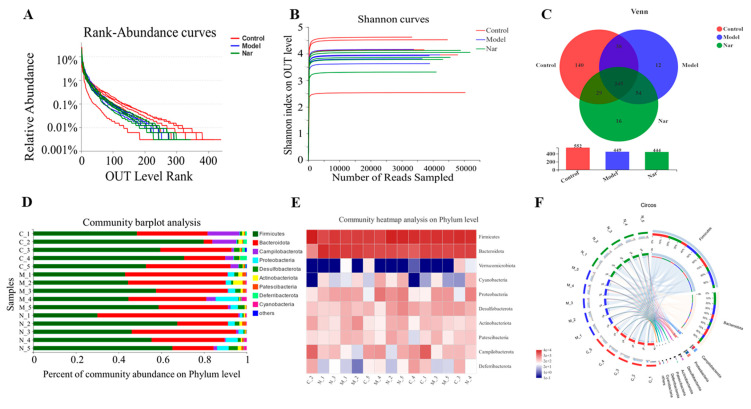
Naringin regulated on the disturbed gut microbiota in DSS-induced colitis mice. (**A**) Rank-Abundance curves of the OUT level. (**B**) Shannon dilution curves. (**C**) Venn diagram of species in the three groups. (**D**) Histogram of intestinal microbial community structure at the phylum level. (**E**) Community heatmap analysis in the phylum levels. (**F**) Circos of samples and species in the phylum levels.

**Figure 7 molecules-26-06604-f007:**
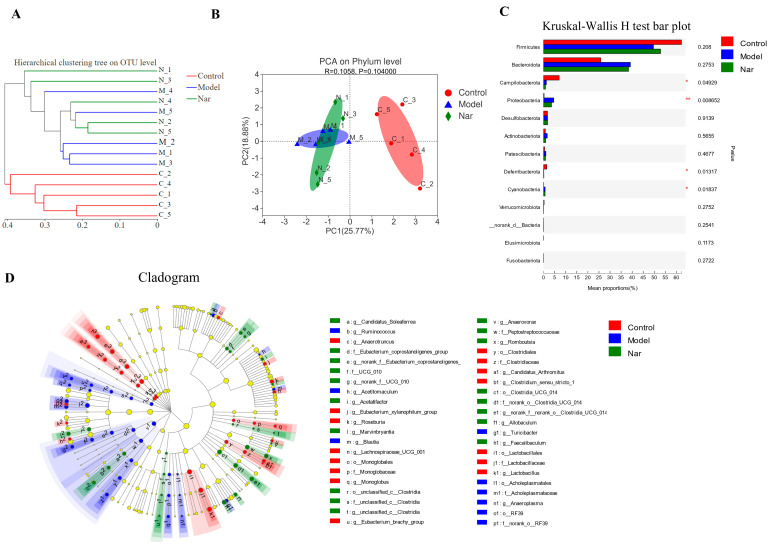
Species difference analysis on the gut microbiota in DSS-induced colitis mice. (**A**) Cluster dendrogram of the three groups on OUT level. (**B**) PCA analysis of variation on phylum level. (*n* = 5). (**C**) The Kruskal-Wallis H test was used to compare the species of intestinal flora in different groups. This figure is the result at the level of phylum classification. * *p* < 0.05 and ** *p* < 0.01, compared with the control group. (**D**) The LEfSe dendrogram showed the phylogenetic distribution of associated colonic microorganisms in three groups of mice.

## Data Availability

Data is contained within the article and Appendix A.

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
