# Peer review of "Naringin Exhibited Therapeutic Effects against DSS-Induced Mice Ulcerative Colitis in Intestinal Barrier–Dependent Manner"

_molecules, 2021, doi:10.3390/molecules26216604_

Round 1

Reviewer 1 Report

The authors have done their study using pure naringin compound- How does the concentration of naringin found in grapefruit compare to the concentration used in this study?

Each section of the results discusses the generic conclusions drawn from DSS groups vs naringin treatment, more specific details about differences seen between various concentrations of naringin, dss groups vs model, dss groups vs control along with statistical significance of these results should be included in this section

Figure 3b and 3c are missing 25 mg/kg treatment results

Were there any previous studies for each of the observed results that support author conclusions?

It is unclear from the discussion how naringin treatment might restore gut microflora as the tables and figures of PCA plots and cluster analysis do not seem to show very less similarities in the microflora of control and naringin group 

Reviewer 2 Report

The manuscript “Naringin exhibited therapeutic effects against DSS-induced 2 mice ulcerative colitis in intestinal barrier–dependent manner” of Ruige Cao et al. is well written and of great of interest. They observed that the administration of polyphenol Naringin in mice affected by ulcerative colitis improved the weight loss, reduced DAI score and alleviated colonic tissue damage, alleviated oxidative stress and reduced the production of inflammatory cytokines, partially restore the intestinal biodiversity and promote the restoration of intestinal microbiota’s structure. Naringin was found to act on mucosal TJ proteins. I recommend this paper for publication.

Reviewer 3 Report

This article presents the therapeutic effects of naringin on DSS-induced mice UC and tries to focus on its mechanism. Based on faithful experiments, this article explains effects of naringin. Unfortunately, the protective effect of naringin of DSS-induced  mice UC, decrease of pro-inflammatory cytokines and upregulation of ZO-1 protein were reported as the authors commented. This important paper was not referred in this article.  Instead, this paper newly explains the intestinal bioflora aspect. I recommend to organize this article to stress new findings and avoid / reduce overlap  with the previous results. 

  • Minor comment : "up-regulation" and upregulation are mixed. Express as one term.  
  •  

Round 2

Reviewer 1 Report

The revised manuscript provides necessary details to be considered for publication.